# Optimal-Damage-Effectiveness Cooperative-Control Strategy for the Pursuit–Evasion Problem with Multiple Guided Missiles

**DOI:** 10.3390/s22239342

**Published:** 2022-11-30

**Authors:** Xiang Ma, Keren Dai, Man Li, Hang Yu, Weichen Shang, Libo Ding, He Zhang, Xiaofeng Wang

**Affiliations:** 1ZNDY of National Defense Key Laboratory, School of Mechanical Engineering, Nanjing University of Science and Technology, Nanjing 210094, China; 2North Information Control Research Academy Group Co., Ltd., Nanjing 211153, China; 3School of Mechatronical Engineering, Beijing Institute of Technology, Beijing 100081, China

**Keywords:** cooperative control, pursuit–evasion problem, guidance-integrated fuze, dynamic systems analysis, adaptive parameter adjusting, virtual force

## Abstract

In this paper, an optimal-damage-effectiveness cooperative-control strategy based on a damage-efficiency model and a virtual-force method is proposed to solve the pursuit–evasion problem with multiple guided missiles. Firstly, different from the overly ideal assumption in the traditional pursuit–evasion problem, an optimization problem that maximizes the damage efficiency is established and solved, making the optimal-damage-effectiveness strategy more meaningful for practical applications. Secondly, a modified virtual-force method is proposed to obtain this optimal-damage-effectiveness control strategy, which solves the numerical solution challenges brought by the high-complexity damage function. Thirdly, adaptive gain is designed in this strategy based on guidance-integrated fuze technology to achieve robust maximum damage efficiency in unpredictable interception conditions. Finally, the effectiveness and robustness of the proposed strategy are verified by numerical simulations.

## 1. Introduction

With the rapid development of cyber–physical systems and complex dynamic networks, cluster confrontation is becoming the mainstream direction in aerospace. In the field of cluster work, cooperative swarm-intelligence decision-making is the key to success in various missions, including UAV group confrontations [1], grouped underwater robot cooperative searching [2], and grouped missile cooperative control [3].

Missile cooperative interception is very much in line with the needs of the field of cluster work, and there are always great hidden dangers in using only one missile for interception. Due to the increasing mobility of targets, the defensive line is very likely to be broken. Grouped missile cooperative interception can greatly improve the success rate of intercepting invading aircraft. The optimal strategy of this pursuit–evasion game is actually an important method for defending important targets [4]. This issue has also attracted worldwide attention. Several studies have focused on multiple low-speed ‘pursuers’ hunting a high-speed ‘evader’ [5,6,7,8]. In [5], the scenario involves a single pursuer and a single evader, the latter of which is slower. Furthermore, refs. [6,7,8] cover the Target–Attacker–Defender scenario where the Attacker seeks to capture the Target, and the Defender seeks to capture the Attacker, thereby protecting the Target. For the case of multiple pursuers, the Hamilton–Jacobi (HJI) equation [9] is applied to achieve cooperative decision-making, but its direct numerical solution is difficult and even unsolvable [10]. Therefore, an approximate numerical solution of the HJI equation is the key to such a problem. Compared with reinforcement learning [11,12] and other methods, the “virtual force” method [13] has unparalleled advantages in computational complexity. This is a strategy based on the position of the two sides and the position of the defense line and is independent of the initial cost function in obtaining the optimal control method. Thus, it is the most practical tool for the solution of the optimal strategy in the cooperative pursuit–evasion problem.

For all these traditional pursuit–evasion games, modeling is based on the concept of the Apollonius circle [14], which essentially sets the successful condition of achieving a final range between the pursuers and evader below a certain threshold as the termination [15]. Typically, the capture time or distance from the target is used as a performance criterion in such scenarios. However, for the actual missile control operation, maximizing the damage effect on the evader is the essential object. Unfortunately, the damage effect is a complex function of multiple factors, including the distance, direction, and relative velocity vector. Therefore, it is necessary to reform the overly ideal Apollonius circle assumption by considering the damage effect, which is determined by the damage-efficiency model [16], to make the optimization strategy more meaningful for practical applications.

Obviously, real-time estimation of the damage effect function relies on a variety of types of data detection by the guidance-integrated fuze system [17,18,19,20]. It is worth noting that the priority of distance and the direction for damage-effect optimization varies during different stages of the pursuit. In the early stage of the pursuit, it is difficult for the fuze to accurately estimate the final intersection direction between the pursuer and the evader, so reducing the distance between the pursuer and the evader as quickly as possible is the dominant purpose. In the later stages of the pursuit, the direction is increasingly important for the final damage effect and should be the dominant factor in the control strategy [3,21,22]. Therefore, the optimal control strategy must dynamically set the priority of distance and direction during the progress of the pursuit.

Based on the above overview, there are still two difficulties to be overcome in cooperative control. To address them, we propose a more practical and easy-to-implement optimal-damage-effectiveness cooperative-control strategy, and the main contributions are as follows:(1)By replacing the overly ideal Apollonius circle concept with the damage-efficiency model, the optimization model of the cooperative pursuit–evasion game is reconstructed to maximize the damage efficiency on the evader.(2)The heuristic “virtual force” method is reformed to be compatible with the damage-efficiency model, enabling a low-complexity numerical solution to the damage-effectiveness cooperative-control strategy.(3)The adaptive gains for distance and direction are elaborately designed in this optimal-damage-effectiveness strategy to address their variable priorities during control, ensuring robust maximum damage efficiency in unpredictable control conditions.

The remainder of this paper is designed as follows: Section 2 formulates the problem, introduces the movement model of guided ammunition, and establishes the damage-efficiency function to determine the end condition. Section 3 designs strategies for pursuers and evaders. In Section 4, a numerical simulation and analysis are carried out. Finally, Section 5 concludes the paper.

## 2. Problem Formulation

### 2.1. Guided Munition Movement Model

The relative motion model of pursuer and evader is shown in Figure 1: $p_{i}\left(i=1,2,\ldots ,n\right)$ represents the pursuers, and all guided missiles are the same type, so the movement model can be described as follows: (1)$${\dot{x}}_{pi}=v_{pi}\cos\left(\theta_{pi}\right){\dot{y}}_{pi}=v_{pi}\sin\left(\theta_{pi}\right){\dot{\theta }}_{pi}=w_{pi}$$
where $x_{pi}$ and $y_{pi}$, respectively, are the positions of the *x* and *y* directions of the *i*-th pursuer; $v_{pi}$ is the speed of the *i*-th pursuer; $\theta_{pi}$ is the course angle of the *i*-th pursuer; and $w_{pi}$ is the input of the *i*-th pursuer. The variable $p_{e}$ represents the evader; its movement model can be described as follows: (2)$${\dot{x}}_{e}=v_{e}\cos\left(\theta_{e}\right){\dot{y}}_{e}=v_{e}\sin\left(\theta_{e}\right){\dot{\theta }}_{e}=w_{e}$$
where $x_{e}$ and $y_{e}$, respectively, are the positions of the *x* and *y* directions of the evader; $v_{e}$ is the speed of the *i*-th evader; $\theta_{e}$ is the course angle of the evader; and $w_{e}$ is the input of the evader.

Based on the above motion model, the following assumptions are made in the interception process:(1)A line of defense is set up at x=0, the pursuer’s mission is to destroy the evader before the evader breaks through the line of defense, and the evader’s mission is to break through the line of defense before it is destroyed, as shown in Figure 1.(2)Due to the limitation of the driving force, the input of the movement mode of the pursuer and the evader is limited to the constants $w_{p\max}$ and $w_{e\max}$. The input of the movement mode of the pursuer also satisfies a certain proportion relationship [23,24]. Because of the above assumption, the pursuer is slower than the evader, i.e.,
(3)$$\lambda =\frac{v_{p}}{v_{e}},\;\lambda <1,\left|w_{pi}\right|\leq w_{p\max},\left|w_{e}\right|\leq w_{e\max}$$(3)The location of the line of defense is determined, as are the number of pursuers and evaders. The evader cannot get the precise initiation range and damage function of the pursuer but only the speed and position information of the pursuer. The pursuer can obtain all state information from the system of the guidance-integrated fuze and can make a real-time decision.

**Remark** **1.**
*For the above assumptions, this model is only applicable to the pursuit–evasion problem with a proportion of fixed speed. Because there are more pursuers, the pursuers have worse performance than the evader. However, the detection system of the pursuer is better than that of the evader under the model proposed in this paper. In addition to position detection of the evader, heading angle detection of the evader is added.*


### 2.2. Damage Efficiency Model

In this problem, not only does the motion control of the pursuer need to be taken into account, but the specific initiation time also needs to be decided. The information is obtained through the guidance system and the fuze system then combined with the damage-efficiency model to judge the damage efficiency under the current state. Therefore, it is changed from the general pursuit–evasion problem, which judges the outcome of both sides of the game based on distance, to the space–time optimization problem that needs to combine the initiation time and damage range. Thus, this paper introduces the damage-efficiency model to further evaluate the damage efficiency. To better measure the damage degree, it is generally considered that the greater the impact kinetic energy of the fragment, the stronger the damage effect [25,26]. Therefore, the damage efficiency function is introduced to determine the degree of destruction of the target, which can be expressed as follows:(4)$$F_{bi}=f_{kd}\cdot {∥v_{dgi}∥}_{2}$$
where $f_{kd}$ is the damage efficiency coefficient, and ${∥v_{dgi}∥}_{2}$ is the square of the fragment impact velocity of the pursuer.

The initial detonation velocity in the warhead can be calculated using the Gurney coefficient [26]:(5)$$v_{0}=\sqrt{2E}\eta \sqrt{\frac{\beta }{1+\frac{\beta }{2}}}$$
where η is the fragment initial-velocity-correction coefficient; $\sqrt{2E}$ is the Gurney coefficient of the explosive; and β is the ratio of charge mass to the warhead. Then, angle $\theta_{f}$ between the fragment velocity vector and the warhead normal is as follows:(6)$$\theta_{f}=\arctan\left(\frac{v_{0}}{2D}\cos\left(x_{m}/R\right)\right)$$
where $\cos\left(x_{m}/R\right)$ is the included angle between the normal line of the shell and the propagation direction of the detonation wave, $x_{m}$ is the abscissa of the pre-supported fragment, *R* is the radius of the warhead, and *D* is the explosive detonation velocity. Thus, the fragment’s scatter angle is in the range of $\theta_{d0}\in \left(\pi /2-\theta_{f},\pi /2+\theta_{f}\right)\cup \left(-\pi /2-\theta_{f},-\pi /2+\theta_{f}\right)$. When the guided ammunition is in motion, as shown in Figure 2, the initial velocity of the fragment is $v_{0}$. Combined with the currently guided ammunition speed $v_{pi}$, the initial static velocity $v_{d0}$ of the fragment in the projectile coordinate system is:(7)$$\begin{array}{cc} \; & v_{d0}=\sqrt{{\left(v_{0}\cos\theta_{d0}+v_{pi}\right)}^{2}+{\left(v_{0}\sin\theta_{d0}\right)}^{2}},\theta_{d0}=\arctan\left(\left(v_{0}\cos\theta_{0}+v_{pi}\right)/v_{0}\sin\theta_{0}\right) \end{array}$$

Due to the influence of air resistance, gravity, and other factors, the fragment velocity will be attenuated, so the actual fragment velocity $v_{d}$ in the body coordinate system is:(8)$$v_{d}=v_{d0}e^{-k_{T}r}$$
where *R* is the fragment flight distance; $k_{r}$ is the attenuation coefficient, which can be expressed as $k_{r}=C_{x}\rho A/2m_{f}$; $C_{x}$ is the resistance coefficient; ρ is the air density; and $A/2m_{f}$ is the ratio of the windward area of the fragment to the fragment mass. The relative motion of the pursuer and evader in the ground coordinate system is shown in Figure 3. Based on the transformation relationship between the body coordinate system and the ground coordinate system, the value $v_{dg}$ of the fragment in the ground coordinate system is:(9)$$\begin{array}{cc} v_{dgi} & =B_{G}\cdot v_{d}\\ & =\left[\begin{array}{cc} \cos\theta_{pi} & -\sin\theta_{pi}\\ \sin\theta_{pi} & \cos\theta_{pi} \end{array}\right]\cdot \left[\begin{array}{c} \left(v_{0}\cos\theta_{0}+v_{pi}\right)e^{-k_{r}r}\\ \left(v_{0}\sin\theta_{0}\right)e^{-k_{r}r} \end{array}\right]\\ & =\left[\begin{array}{c} \cos\theta_{pi}\left(v_{0}\cos\theta_{0}+v_{pi}\right)e^{-k_{r}r}-\sin\theta_{pi}\left(v_{0}\sin\theta_{0}\right)e^{-k_{r}r}\\ \sin\theta_{pi}\left(v_{0}\cos\theta_{0}+v_{pi}\right)e^{-k_{r}r}+\cos\theta_{pi}\left(v_{0}\sin\theta_{0}\right)e^{-k_{r}r} \end{array}\right] \end{array}$$
where $B_{G}$ is the coordinate transformation matrix. The square of the fragment impact velocity of the pursuer can be expressed as follows:(10)$$\begin{array}{cc} {∥v_{dgi}∥}_{2} & ={∥v_{e}-v_{dgi}∥}_{2}={∥\begin{array}{c} v_{e}\cos\theta_{e}-\left(\cos\theta_{pi}\left(v_{0}\cos\theta_{0}+v_{pi}\right)e^{-k_{p}r}\right.\\ \left.-\sin\theta_{pi}\left(v_{0}\sin\theta_{0}\right)e^{-k_{p}r}\right)\\ v_{e}\sin\theta_{e}-\left(\sin\theta_{pi}\left(v_{0}\cos\theta_{0}+v_{pi}\right)e^{-k_{p}r}\right.\\ \left.+\cos\theta_{pi}\left(v_{0}\sin\theta_{0}\right)e^{-k_{p}r}\right) \end{array}∥}_{2} \end{array}$$

Due to the scattering characteristics of fragments, more fragments with high kinetic energy will gather in the middle line of the damage range; thus, the expression of the damage efficiency coefficient is as follows:(11)$$f_{kd}=\max\left\{\frac{k_{d}}{\sqrt{2\pi }\sigma }e^{-\frac{{\left(\theta_{0}-\pi /2-\theta_{pi}\right)}^{2}}{2\sigma^{2}}},\frac{k_{d}}{\sqrt{2\pi }\sigma }e^{-\frac{{\left(\theta_{0}+\pi /2-\theta_{pi}\right)}^{2}}{2\sigma^{2}}}\right\}$$
where $k_{d}$ is the gain of damage efficiency, and σ is the standard deviation of damage efficiency. The damage efficiency satisfies a Gaussian distribution. In the projectile coordinate system, the closer the relative positions between projectile target positions are to ±π/2, the greater the damage efficiency is.

## 3. Pursuit–Evasion Strategy

Based on the above description, the following coordination variables are designed to better solve the above problems:(12)$$\left\{\begin{array}{c} r_{pi}=\sqrt{{\left(x_{pi}-x_{e}\right)}^{2}+{\left(y_{pi}-y_{e}\right)}^{2}}\\ r_{e}=\sqrt{{\left(x_{e}-x_{t}\right)}^{2}} \end{array}\right.$$
where $r_{pi}$ is the Euclidean distance between the *i*-th pursuer and the evader, $r_{e}$ is the Euclidean distance between the evader and the line of defense, and $x_{t}$ is the x-coordinate of the line of defense. The conditions of the end decision are as follows:(1)The evader breaks through the line of defense without being destroyed by the pursuer, which is $x_{t}\leq 0$.(2)The pursuer meets the initiation condition and causes certain damage to the evader.

### 3.1. Evader Strategy

For the evader, since the specific damage-efficiency model of the guided missiles cannot be accurately obtained, the evader strategy design scheme can follow the design method of the general pursuit–evasion model. Before designing the evader strategy, we can analyze the situation of the pursuer first [13]. The cost function corresponding to the design of pursuers is made as follows:(13)$$J_{i}=\frac{1}{2}\left(k_{epi}r_{pi}^{2}-k_{e}r_{e}^{2}\right)\;i=1,\ldots ,n$$
where $k_{epi}$ is the gain of evasion, and $k_{e}$ is the gain of breakthrough. Take the derivative of Equation (Equation 13) to obtain:(14)$$\begin{array}{cc} {\dot{J}}_{i} & =k_{epi}\left[\left(x_{pi}-x_{e}\right)\left({\dot{x}}_{pi}-{\dot{x}}_{e}\right)+\left(y_{pi}-y_{e}\right)\left({\dot{y}}_{pi}-{\dot{y}}_{e}\right)\right]-k_{e}\left[\left(x_{e}-x_{t}\right){\dot{x}}_{e}+\left(y_{e}-y_{t}\right){\dot{y}}_{e}\right]\\ \; & =k_{epi}v_{pi}\left[\left(x_{pi}-x_{e}\right)\cos\theta_{pi}+\left(y_{pi}-y_{e}\right)\sin\theta_{pi}\right]\\ \; & -v_{e}\left[\left(k_{e}x_{e}-k_{e}x_{t}-k_{pi}x_{pi}+k_{pi}x_{e}\right)\cos\theta_{e}\right.\left.+\left(k_{e}y_{e}-k_{e}y_{t}-k_{pi}y_{pi}+k_{pi}y_{e}\right)\sin\theta_{e}\right] \end{array}$$

It can be concluded from Equation (Equation 14) that no matter what kind of evader strategy is adopted, if the pursuer $p_{i}$ needs to shorten the distance $r_{pi}$, it can adopt the following strategy to minimize the cost function $J_{i}$; the $\theta_{pi}^{*}$ can be expressed as:(15)$$\begin{array}{c} \sin\theta_{pi}^{*}=-\frac{x_{pi}-x_{e}}{\sqrt{{\left(x_{pi}-x_{e}\right)}^{2}+{\left(y_{pi}-y_{e}\right)}^{2}}},\cos\theta_{pi}^{*}=\frac{y_{pi}-y_{e}}{\sqrt{{\left(x_{pi}-x_{e}\right)}^{2}+{\left(y_{pi}-y_{e}\right)}^{2}}} \end{array}$$

According to Equation (Equation 15), the input of $p_{i}$ can be obtained as follows:(16)$$w^{*}=\theta_{pi}^{*}-\theta_{pi}$$

Since the finite maneuver condition of Assumption (2) means that the control input is bounded, the input needs to satisfy:(17)$$w_{pi}=\left\{\begin{array}{cc} w_{p\max} & w_{pi}^{*}>w_{p\max}\\ w_{pi}^{*} & \left|w_{pi}^{*}\right|\leq w_{p\max}\\ -w_{p\max} & w_{pi}^{*}<-w_{p\max} \end{array}\right.$$

However, for the evader, it is impossible to achieve the maximum value at the same time by solving Equation (Equation 13) directly. Although the optimal solution of a certain $J_{i}\left(i=1,\ldots ,n\right)$ can be solved through Equation (Equation 14), due to the different states (i.e., positions or heading angles) of the pursuers, each pursuer also has a different effect on the evader. Obviously, the evader cannot simultaneously maximize each pursuer’s cost. So the following two principles should be met based on the characteristics of the evader.
(1)Breakthrough principle: the distance $r_{e}$ between the evader and the line of defense should be narrowed to as small as possible. This requires the evader to be able to break through the defense line as soon as possible, thus optimizing Function (18).(2)Avoidance principle: the value of $r_{pi}$ is as large as possible. This requires the evader to stay away from the pursuers to prevent it from being destroyed by the pursuers, thus also optimizing Function (18).

Based on two principles above, the cost function of the evader can be expressed as:(18)$$J_{e}=\frac{1}{2}\left(k_{e}r_{e}^{2}-k_{epi}r_{pi}^{2}\right)\;i=1,\ldots ,n$$
where it is very difficult for the evader strategy to satisfy both principles at the same time, especially in the case of multiple pursuers; therefore, the weight of each principle needs to be adjusted based on different situations. Combined with the concept of virtual force in the literature [13], we can think that each pursuer and evader is a mass, and different pursuers will have a virtual force on the evader, which can be divided into repulsion under threat and gravity that needs to break through the defense line, as shown in Figure 4. The particle *e* will be subjected to the gravitational force $F_{0}$ from the line of defense due to Principle (1); the particle *e* receives repulsive force $F_{i}\left(i=1,\ldots ,n\right)$ from particle $p_{i}\left(i=1,\ldots ,n\right)$ due to Principle (2).

To solve the cost function (18) more accurately, an adaptive-gain virtual-force method is designed considering the threat degree of different pursuers based on [13]. The farther the distance is, the smaller the repulsive force of the pursuer on the evader is; otherwise, larger than the repulsive force is the gravitational force of the defense line to the pursuer. The threat of the pursuer is lower and the gravitational force is greater while the distance between the pursuer and the evader is farther. Thus, the repulsive forces $F_{i}\left(i=1,\ldots ,n\right)$ and $F_{0}$ can be designed as follows:(19)$$F_{i}=k_{epi}/r_{pi}^{2},F_{0}=k_{e}/\left(\sum_{i=1}^{n}k_{epi}/r_{pi}^{2}\right)$$

Based on the above analysis, the adaptive laws of $k_{epi}$ and $k_{e}$ in Function (19) are as follows:(20)$$k_{epi}=\sum_{i=1}^{n}r_{pi}/r_{pi}^{2},k_{e}=1/\left(\sum_{i=1}^{n}k_{epi}\right)$$

Therefore, we can calculate the resultant force of such a pursuit model to obtain the direction of the evader. The virtual resultant forces in the horizontal direction and the vertical direction can be expressed as:(21)$$\begin{array}{cc} \; & F_{ex}=F_{0}+\sum_{i=1}^{n}F_{i}\cos\alpha_{i},F_{ey}=\sum_{i=1}^{n}F_{i}\sin\alpha_{i}\\ \; & \sin\alpha_{i}=\left(x_{e}-x_{pi}\right)/r_{pi},\cos\alpha_{i}=\left(y_{e}-y_{pi}\right)/r_{pi} \end{array}$$

Based on the above analysis, the direction of the resultant force and input $w_{e}$ can be obtained as:(22)$$\begin{array}{cc} \; & \sin\theta_{e}^{*}=F_{ey}/\sqrt{F_{ex}^{2}+F_{ey}^{2}},\cos\theta_{e}^{*}=F_{ex}/\sqrt{F_{ex}^{2}+F_{ey}^{2}},w_{e}^{*}=\theta_{e}^{*}-\theta_{e} \end{array}$$

Since the finite maneuver condition of Assumption (2) means that the control input is bounded, the input needs to satisfy:(23)$$w_{e}=\left\{\begin{array}{cc} w_{e\max} & w_{e}^{*}>w_{e\max}\\ w_{e}^{*} & \left|w_{e}^{*}\right|\leq w_{e\max}\\ -w_{e\max} & w_{e}^{*}<-w_{e\max} \end{array}\right.$$

### 3.2. Pursuer Strategy

Since the pursuer’s goal is to destroy the evader, the effect of the damage function also needs to be considered. Therefore, it is impossible to design the cost function according to the general pursuit–evasion problem; thus, the cost function can be written according to the design method of the damage function combined with Equation (Equation 4):(24)$$J_{p}=1/\sum_{i=1}^{n}F_{bi}\left(r_{pi},v_{pi},v_{e},\theta_{pi},\theta_{e}\right)\;i=1,\ldots ,n$$
where $F_{bi}$ represents the damage to the evader caused by the *i*-th pursuer detonating at its current position. The damage range of the pursuer is shown in Figure 5. In combination with Equation (Equation 4), it is expressed as follows:(25)$$\theta_{pi}^{*}=\arg\min1/(f_{kd}\cdot \sum_{i=1}^{n}∥v_{dgi}{∥}_{2})$$
where $\theta_{pi}^{*}=\left[\theta_{p1}^{*},\theta_{p2}^{*},\ldots ,\theta_{pn}^{*}\right]$ represents the cost function solution. Since $\theta_{0}$ represents the fragment scatter angle, the $\theta_{0}$ angle fragments that can cause damage to the target are determined by the relative position of the pursuer and evader at the initiation time. Thus, it is impossible to determine the specific value of $\theta_{0}$ in the movement stage of the pursuer. Therefore, the relative position of the pursuer and evader is used to solve for $\theta_{0}$ to solve this problem. However, this solution method may cause a certain risk in the initiation of the pursuer: that is, the evader does not fall in the damage range or falls on the edge of the damage range. In the actual situation, the distribution of damage efficiency satisfies a normal distribution, and the center of the sector is the central value of the normal distribution: that is, more fragments with high damage efficiency will fall near the middle line of the damage range. Because of this risk, the resulting value may lead to a decrease in damage efficiency. In addition, the calculation of the minimum value of the above functions by using the interior-point method is very large, and the accuracy of the results will be affected if the accuracy and step-size of the solution are adjusted.

Based on the above points, it is very difficult to directly solve the problem of the cost Function (25). Therefore, three principles are proposed for the cost Function (25), and a virtual-force-analysis method based on these principles is established to approximate the solution.
(1)Capture principle: the distance $r_{pi}$ between the pursuer and the evader should be narrowed as quickly as possible. This not only helps to achieve interception as soon as possible but also helps to increase the fragment kinetic energy relative to the evader at the initiation time, thus optimizing Function (21).(2)Azimuth correction principle: in terms of azimuth, the pursuer tries to keep the evader in the center of the damage area. This is conducive to the interception of more fragments with high kinetic energy that can act on the evader at the initiation moment, thus optimizing Function (25).(3)Coordinated initiation principle [27,28,29]: if pursuers are required to attack the evader simultaneously, the remaining time from each pursuer to the evader is required to be the same for all pursuers. Since each pursuer has the same velocity, the distance $r_{pi}$ between the pursuers and the evader should be as equal as possible. This is conducive to the fact that the damage ranges of multiple pursuers can act on the evader at the same time to achieve the superposition of damage effectiveness to achieve the purpose of high damage effectiveness.

Based on the three principles above, the approximate cost function is derived based on the virtual-force method. For the fast-capture principle, the process of analysis is similar to Equations (12)–(16). The distance cost function can be directly set as:(26)$$F_{pi}=r_{pi}^{2}$$

For the azimuth correction principle, since it is very difficult to solve $v_{dgi}$, it is necessary to determine the characteristic conditions of initiation and optimize the function to achieve the purpose of maximum damage and accurate initiation. Combined with the damage-efficiency function, it is required that the evader is within the range of initiation and its speed direction is opposite to the flight direction of the fragment. In other words, the center-line of the dynamic flight angle of detonation at the moment is in a straight line with the heading angle of the evader to meet the maximum damage condition and reduce the probability of escape; the damage range is shown in Figure 5. Then, the initiation pose cost function can be reconstructed as follows:(27)$$\begin{array}{cc} F_{di} & =\min\left\{{\left(\theta_{pi}+\theta_{psi}-\theta_{e}^{\prime }\right)}^{2},{\left(\theta_{pi}-\theta_{psi}-\theta_{e}^{\prime }\right)}^{2}\right\}/r_{pi}^{2}\\ \theta_{e}^{\prime } & =\left\{\begin{array}{cc} \theta_{e}-\pi & \theta_{e}\geq 0\\ \theta_{e}+\pi & -2\pi <\theta_{e}<0\\ \theta_{e}+3\pi & \theta_{e}\leq 0 \end{array}\right. \end{array}$$
where $\theta_{psi}$ is the center-line of the dynamic scattering angle of the *i*-th pursuer; $\theta_{e}^{\prime }$ is the opposite of the evader’s motion. Since an initiation angle needs to be aligned, it can be considered that the particle $p_{i}$ is subjected to a torque $F_{di}$ for angular alignment.

The cooperative principle requires the pursuers to attack the target simultaneously so as to increase the efficiency of damage. The cooperative cost function is defined as:(28)$$F_{ci}={\left(r_{pi}-\sum_{i=1}^{n}r_{pi}/n\right)}^{2}$$
where in Equation (Equation 28), the cooperativity of the *i*-th pursuer can be described. The higher the value of $|F_{c}i|$ is, the worse the cooperativity. Then, the cooperativity can be adjusted through the control input to achieve the cooperative attack of pursuers. If $r_{pi}\geq \sum_{i=1}^{n}r_{pi}$, particle $p_{i}$ receives a gravitational force towards particle *e* to rapidly shorten its distance from particle *e*; if $r_{pi}<\sum_{i=1}^{n}r_{pi}$, particle $p_{i}$ receives a repulsive force in the opposite direction of *e*. The virtual-force analysis of the pursuer based on the above design principles is shown in Figure 6.

The pursuer’s resultant force is as follows:(29)$$F_{pi}^{\prime }=k_{pi}F_{pi}+k_{di}F_{di}+k_{ci}F_{ci}$$
where $k_{pi}$ is the gain of pursuit, $k_{di}$ is the gain of damage, and $k_{ci}$ is the gain of cooperativeness. It is worth noting that for guided ammunition controlled by a guidance-integrated fuze, the interception process goes through the stages of long-range guidance control and short-range fuze control. In this process, the dominant position of each gain in Function (29) also changes. Therefore, we propose an interception-state-adaptive variable-weight design for Function (29) so that its solution can better approximate the solution of Function (24). Specifically, in the stage of long-range guidance control, $k_{pi}$ is increased to quickly approach the evader when the distance between the *i*-th pursuer and the evader is far. In the stage of short-range fuze control, $k_{di}$ is increased to ensure a proper initiation azimuth and to maximize damage efficiency when close to the evader. The factor $k_{ci}$ ensures coordinated initiation of different pursuers during a cooperative interception. Based on the above analysis, the adaptive laws of $k_{pi}$, $k_{di}$, and $k_{ci}$ in Function (29) are as follows:(30)$$\begin{array}{c} k_{pi}=r_{pi}^{2},k_{di}=\sum_{i=1}^{n}r_{pi}/r_{pi}^{2},k_{ci}={\left(r_{pi}-\sum_{i=1}^{n}r_{pi}/n\right)}^{2} \end{array}$$

Therefore, the direction of the virtual resultant force on the pursuer can be calculated. If $r_{pi}\geq \sum_{i=1}^{n}r_{pi}$, The resultant virtual force in the horizontal direction and the vertical direction can be divided into the following two cases:(31)$$\left\{\begin{array}{c} F_{pxi}^{\prime }=k_{pi}F_{pi}\cos\alpha_{i}+k_{di}F_{di}\cos\left(\alpha_{i}+\pi /2\right)\\ +k_{ci}F_{ci}\cos\left(\alpha_{i}+\pi \right)\\ F_{pyi}^{\prime }=k_{pi}F_{pi}\sin\alpha_{i}+k_{di}F_{di}\sin\left(\alpha_{i}+\pi /2\right)\\ +k_{ci}F_{ci}\sin\left(\alpha_{i}+\pi \right) \end{array}\left|de^{+}\right|\geq \left|de^{-}\right|\right.$$
(32)$$\left\{\begin{array}{c} F_{pxi}^{\prime }=k_{pi}F_{pi}\cos\alpha_{i}+k_{di}F_{di}\cos\left(\alpha_{i}-\pi /2\right)\\ +k_{ci}F_{ci}\cos\left(\alpha_{i}+\pi \right)\\ F_{pyi}^{\prime }=k_{pi}F_{pi}\sin\alpha_{i}+k_{di}F_{di}\sin\left(\alpha_{i}-\pi /2\right)\\ +k_{ci}F_{ci}\sin\left(\alpha_{i}+\pi \right) \end{array}\left|de^{+}\right|<\left|de^{-}\right|\right.$$
where $de^{+}=\theta_{pi}+\theta_{psi}-\theta_{e}^{\prime }$, $de^{-}=\theta_{pi}-\theta_{psi}-\theta_{e}^{\prime }$. Otherwise $r_{pi}<\sum_{i=1}^{n}r_{pi}$. The virtual resultant force in the horizontal direction and the vertical direction can be expressed as:(33)$$\left\{\begin{array}{c} F_{pxi}^{\prime }=k_{pi}F_{pi}\cos\alpha_{i}+k_{di}F_{di}\cos\left(\alpha_{i}+\pi /2\right)\\ +k_{ci}F_{ci}\cos\alpha_{i}\\ F_{pyi}^{\prime }=k_{pi}F_{pi}\sin\alpha_{i}+k_{di}F_{di}\sin\left(\alpha_{i}+\pi /2\right)\\ +k_{ci}F_{ci}\sin\alpha_{i} \end{array}\left|de^{+}\right|\geq \left|de^{-}\right|\right.$$
(34)$$\left\{\begin{array}{c} F_{pxi}^{\prime }=k_{pi}F_{pi}\cos\alpha_{i}+k_{di}F_{di}\cos\left(\alpha_{i}-\pi /2\right)\\ +k_{ci}F_{ci}\cos\alpha_{i}\\ F_{pyi}^{\prime }=k_{pi}F_{pi}\sin\alpha_{i}+k_{di}F_{di}\sin\left(\alpha_{i}-\pi /2\right)\\ +k_{ci}F_{ci}\sin\alpha_{i} \end{array}\left|de^{+}\right|<\left|de^{-}\right|\right.$$

Based on the above analysis, the direction of the resultant force can be obtained as:(35)$$\begin{array}{c} \sin\theta_{pi}^{*\prime }=F_{pyi}^{\prime }/\sqrt{{F_{pyi}^{\prime }}^{2}+{F_{pyi}^{\prime }}^{2}},\cos\theta_{pi}^{*\prime }=F_{pxi}^{\prime }/\sqrt{{F_{pxi}^{\prime }}^{2}+{F_{pyi}^{\prime }}^{2}},w_{pi}^{*\prime }=\theta_{pi}^{*\prime }-\theta_{pi} \end{array}$$

This takes into account the position information of the evader so that the evader is more likely to be in the damage area on one side of the pursuers. Thus, the input can be expressed as:(36)$$w_{pi}^{\prime }=\left\{\begin{array}{cc} w_{p\max} & w_{pi}^{*\prime }>w_{p\max}\\ w_{pi}^{*} & \left|w_{pi}^{*\prime }\right|\leq w_{p\max}\\ -w_{p\max} & w_{pi}^{*\prime }<-w_{p\max} \end{array}\right.$$

### 3.3. Condition of the Explosive

In the traditional pursuit–evasion problem, the pursuer is considered to have completed the task by reaching the range of the Apollonius circle between the evader and the pursuers [6,8,10]. General fuze designs also use the distance from the target as the basis of initiation. Therefore, both the traditional fuze and the judgment condition for the end of the pursuit–evation depend on the distance: that is, $r_{i}<d$. However, because the initiation time of the fuze is a time and space judgment process integrating various physical information, it is insufficient to judge it by distance only. Therefore, Equation (Equation 9) can be used to determine the best initiation timing based on the information of the fuze and the target. The determination conditions of initiation conditions are as follows: $F_{bi}>F_{set}$. $F_{set}$ is the condition satisfied by the initiation function, and its specific parameters is designed to meet the empirical value of the damage condition, which changes according to different objectives.

## 4. Numerical Simulations

### 4.1. Parameter Setting

As an example, we provide a scene consisting of four pursuers and one evader. The position of the defensive line is a straight line with a coordinate of zero. The initial position design of the pursuers and the evader are shown in Table 1:

Parameters settings of the fuze–warhead matching mode in the numerical simulation model are shown in Table 2:

### 4.2. Simulation Analysis of Pursuit–Evasion Problem

In this section, a mathematical simulation is taken as an example to explicate the effectiveness of the proposed strategy, and the comparison strategies are proposed in [13] and solve Equation (Equation 25) directly. The simulation step size is 0.005. The trajectory and damage range of the pursuers/evader adopting the proposed pursuer strategy are shown in Figure 7a, and the comparison is shown in Figure 7b. It can be seen from Figure 7a that the strategy takes into account the cooperative characteristics of multiple pursuers and the impact of detonation position on the damage so that the damage range overlaps and the damage direction is corrected. At the time of initiation, $p_{1}$ is corrected by 38.2°, and $p_{2}$ is corrected by 11.9°; therefore, the damage degree of evaders is increased based on Principles (2) and (3). It can be seen from Figure 7b that $p_{1}$ and $p_{2}$ successfully detonate and cause certain damage to the evader. However, since the judgment of explosive conditions does not involve cooperative and pose characteristics, the degree of damage to the pursuers is weakened to some extent.

For the strategy of directly solving Equation (Equation 25), due to accuracy of both the simulation and the algorithm having great influence on the solution result, the results vary greatly under different simulation and algorithm solution accuracies. Figure 7c shows the results with simulation step-size of 0.001. The maximum calculation time of the interior-point method solution is $1\times 10^{5}$, and the tolerance is $1\times 10^{-10}$. It can be seen from the figure that its damage range also overlaps, but the overlap mode is different from that of the proposed strategy. However, when the simulation step-size is 0.5, as shown in Figure 7d, the maximum calculation time of the interior-point-method solution is $1\times 10^{3}$, and the tolerance is $1\times 10^{-7}$; only one pursuer can cause effective damage, which greatly reduces the damage efficiency.

Damage efficiency is determined by Equation (Equation 9). The unit fragment damage efficiency of the proposed pursuer strategy is shown in Figure 8a, and the comparison is shown in Figure 8b. The proposed pursuer strategy has significantly improved the damage efficiency of the unit fragment compared with the ordinary situation because of the consideration of the cooperative position and pose information, and the damage efficiency of the cooperative strategy proposed in this paper is $1.612\times 10^{6}$ J at the initiation time. However, in comparison with the literature [13], due to adjustment of the final initiation position and pose, coordinated initiation cannot be achieved, and the damage efficiency is $1.113\times 10^{6}$ J. The proposed strategy improves the damage efficiency by 44.8% compared with the strategy of [13]. Based on the numerical value analysis above, the proposed adaptive-gain method based on different principles of virtual force and guidance-integrated fuze significantly improves the damage efficiency.

For a strategy that directly solves Equation (Equation 25). The simulation step size is 0.0005, the maximum calculation time for the interior-point-method solution is 10,000, and the tolerance is $1\times 10^{-7}$; the results are shown in Figure 8c. It can be seen from this figure that its damage efficiency is $1.661\times 10^{6}$ J, which is higher than the proposed strategy using virtual force. However, the simulation step size is 0.05, the maximum number of calculations is 1000, and the tolerance is $1\times 10^{-2}$, as shown in Figure 8d. The damage efficiency is reduced to $1.337\times 10^{6}$ J. Moreover, under the condition of high precision and the same simulation step size, the solving time becomes 13.4 times that of the proposed strategy. Even under low-accuracy conditions, it is 6.1 times more than the proposed strategy.

In order to explain the advantages and disadvantages of the two methods more clearly, Figure 9 shows the time-cost as the X-axis and the damage efficiency as the Y-axis. As can be seen from Figure 9, due to the high complexity of the direct solution method, its damage efficiency is greater when the calculation accuracy is higher, but it also consumes more computing resources, which is not conducive to direct application in the actual system. If the solution accuracy continues to be reduced, the solution value does not converge. However, the virtual-force method is relatively stable as a whole and is not affected by accuracy, which saves computing resources and is more conducive to use in the actual system. The proposed algorithm can have relatively stable performance even in the case of low computational resource consumption.

Different gains for the pursuer in the proposed pursuer strategy are shown in Figure 10. It can be seen from Figure 10 that the gain changes for the pursuer can be divided into the guidance stage and the fuze stage. Taking $p_{1}$ as an illustration, the proportion of $k_{p1}$ is the largest when the distance $r_{p1}$ is relatively far from 0–4 s. At this time, the main task of $p_{1}$ is to approach the evader as quickly as possible. When the pursuer is in the main function stage of the fuze at 4–4.3 s, the distance $r_{p1}$ is relatively close; then, the gain in $k_{b1}$ would be rapid to adjust for the best detonation position to maximize damage efficiency. Further, $k_{ci}$ also increases to achieve cooperative initiation when $r_{p1}$ deviates far from the mean of $r_{pi}\left(i=1,\ldots ,4\right)$.

### 4.3. Monte Carlo Analysis of Pursuit–Evasion Problem

In practice, the relative positions of the pursuer, the evader, and the line of defense are uncertain. Therefore, it is necessary to consider the ultimate damage effectiveness under different circumstances. The initial range of the evader is shown in Figure 11. The final damage efficiency of the proposed strategy is compared with that of the strategy in [13].

The damage efficiency of the proposed pursuer strategy is shown in Figure 12. The comparison is shown in Figure 13. The X-axis direction range of the breakout adopted is (−500, 2500), the Y-axis range is (3000, 7000), and the Z-axis is the damage efficiency. The average damage efficiency of the proposed strategy is $1.363\times 10^{6}$ J, while that of the compared strategy is $8.078\times 10^{5}$J, which improves the damage efficiency by 68.3%. If an area greater than $6\times 10^{5}$ J is defined as the effective damage area, the effective damage area of the proposed strategy is 89.3%, while that of the compared strategy is 63.2%, which is an improvement of 41.2%. The histogram of the proportional distribution of damage efficiency is shown in Figure 14, from which it can be seen that the proportion of the optimal damage efficiency distribution of the proposed strategy is much larger than that of the compared strategy.

In order to show the improvement in the final damage efficiency of the different design principles, the strategy in [13] is combined with Principle (2) and Principle (3) to carry out simulations using the same initial conditions, and the results are shown in Figure 15 and Figure 16, respectively. The cost functions of the strategy in [13] combined with Principle (2) are as follows:(37)$$F_{pi}^{12}=k_{pi}F_{pi}+k_{di}F_{di}$$

It can be seen from Figure 15 that since Principle (2) has a great influence on the position and pose of detonation, damage efficiency is improved as much as possible. The average damage efficiency can reach $1.0691\times 10^{6}$ J, but the effective damage area is not greatly improved. The cost functions of the strategy in [13] combined with Principle (3) are as follows:(38)$$F_{pi}^{13}=k_{pi}F_{pi}+k_{ci}F_{ci}$$

It can be seen from Figure 16 that the average damage efficiency of the strategy from [13] combined with Principle (3) can reach $1.1008\times 10^{6}$, and the effective damage range can be improved due to its cooperative characteristics so as to ensure the reliability of simultaneous attack.

Monte Carlo simulation is performed for different pursuer speeds, and the simulation speed is designed to be 500–750 m/s. The results are shown in Figure 17 according to the average damage efficiency of the unit fragments and the percentage of effective damage range as the comparison standard. The damage range is more easily satisfied when the speed of the catcher is close to the speed of the breaker, so the damage range adopted by the two principles will gradually converge with increased speed. The results show that the unit fragment damage efficiency of the proposed algorithm increases by 52.62% and the effective damage range increases by 10.69% at different speeds.

## 5. Conclusions

In this paper, a cooperative control strategy considering damage range is designed to solve the problem of cooperative pursuit–evasion of multiple guided missiles. Firstly, the damage-efficiency function is established by using the fuse–warhead matching model, which can maximize the damage efficiency. Furthermore, a kind of pursuit–evasion problem considering damage range is reconstructed by combining the damage efficiency function. Secondly, the strategies of the pursuers and the evader are designed based on the virtual force and a simplified damage-efficiency function, which greatly reduces the computational complexity and is easier to use in an actual system. Thirdly, a parameter-adaptive method based on the main task stage is designed to realize the optimal damage-efficiency strategy, which can make the corresponding different gain self-adaptive so as to achieve the maximum damage efficiency in unpredictable situations. Finally, numerical simulations are performed on fixed occasions and multi-case simulations with different initial positions and different speeds. The results show that the proposed strategy not only improves the average damage efficiency by 68.3%, but it also expands the range of average effective damage for various initial positions by 41.2% over existing approaches in the literature.

## Figures and Tables

**Figure 1 sensors-22-09342-f001:**
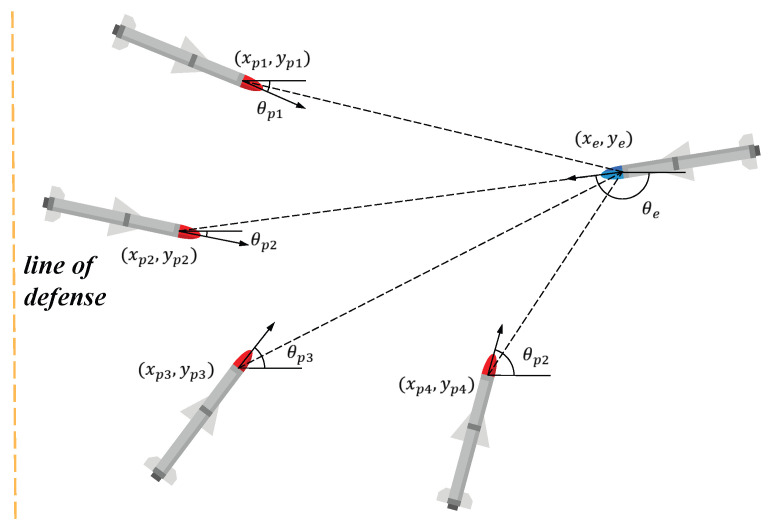
Relative motion.

**Figure 2 sensors-22-09342-f002:**
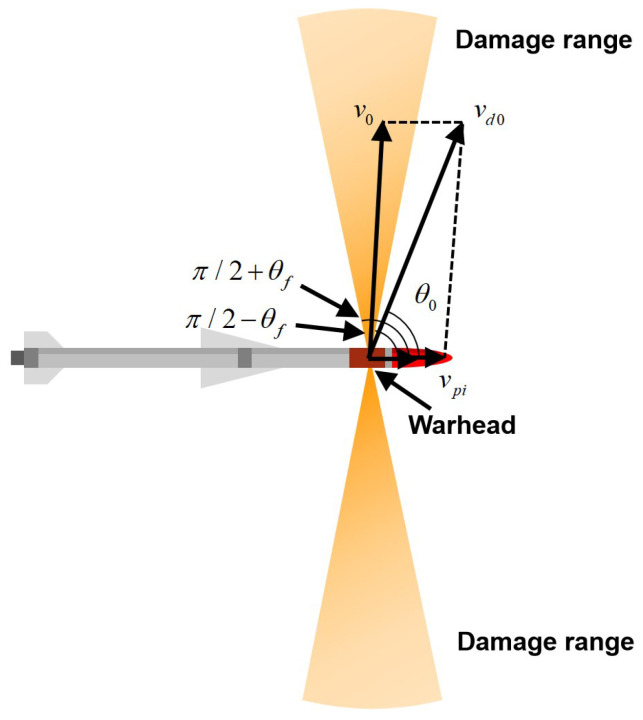
Initial static velocity $v_{d0}$ of the fragment in the projectile coordinate system.

**Figure 3 sensors-22-09342-f003:**
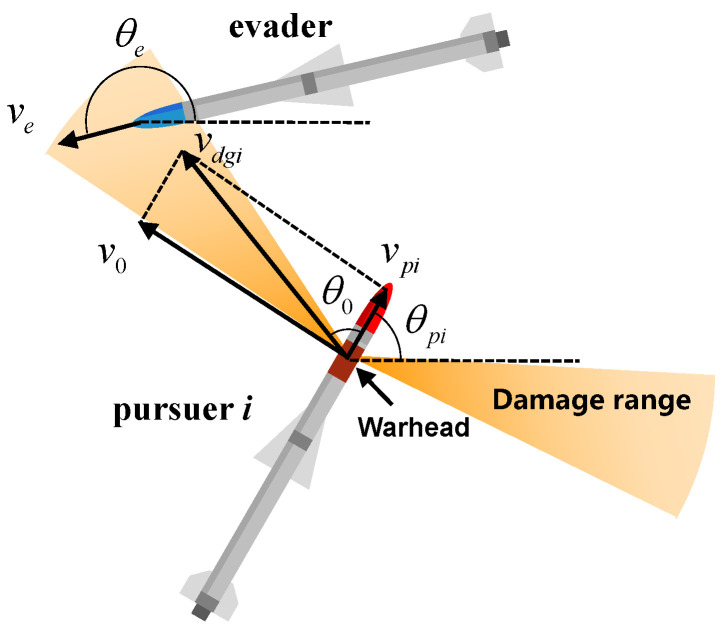
The relative motion of pursuer and evader in the ground coordinate system.

**Figure 4 sensors-22-09342-f004:**
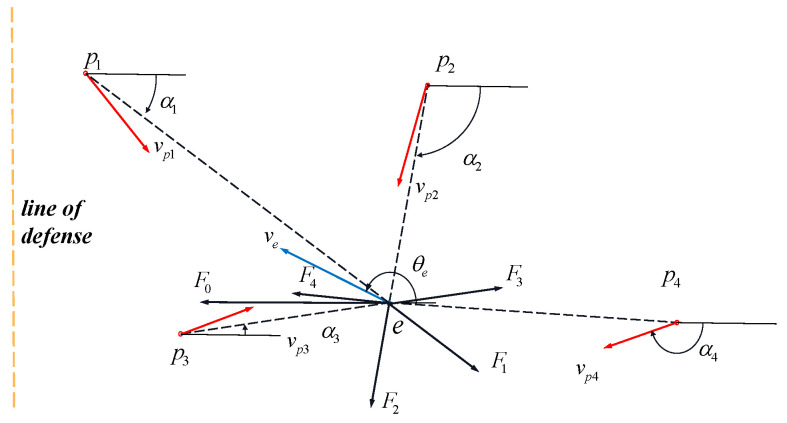
Virtual-force analysis of evaders.

**Figure 5 sensors-22-09342-f005:**
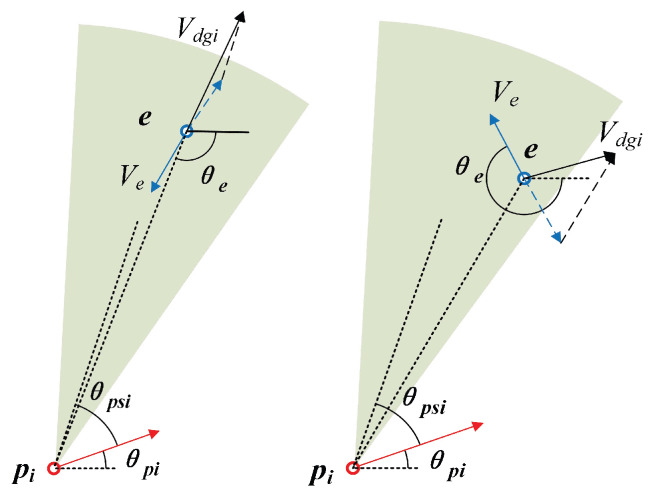
Damage range and movement diagram of the evader.

**Figure 6 sensors-22-09342-f006:**
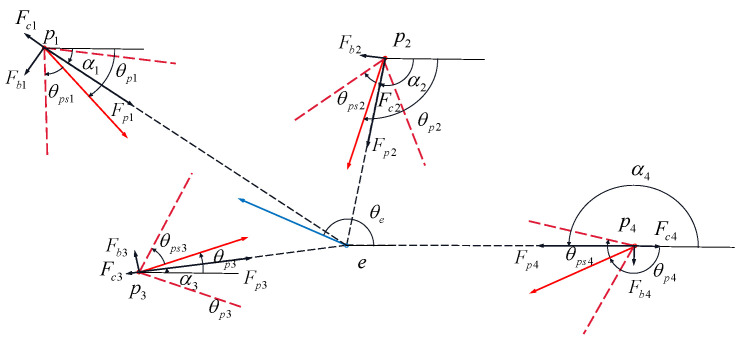
Virtual-force-analysis diagram of the *i*-th pursuer.

**Figure 7 sensors-22-09342-f007:**
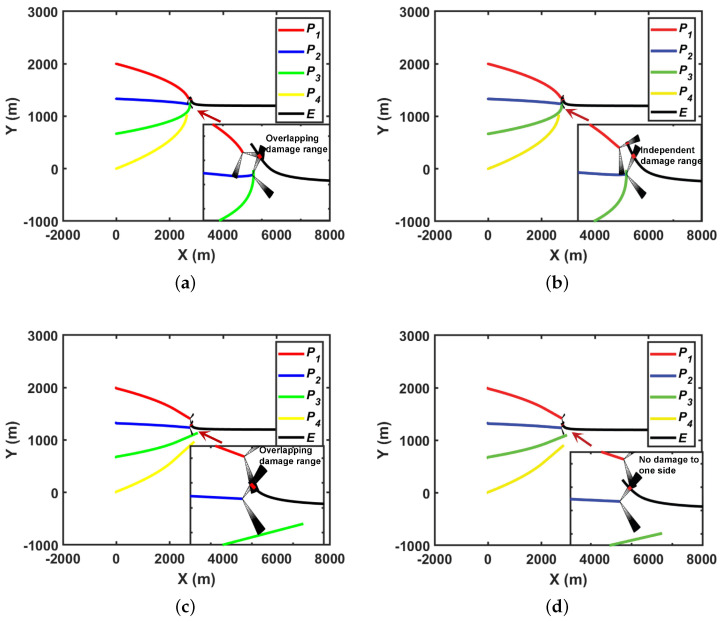
Trajectory and damage range of pursuers/evader: (**a**) proposed strategy; (**b**) strategy from the literature [13]; (**c**) strategy of Equation (Equation 24) with high precision; and (**d**) strategy of Equation (Equation 24) with low precision.

**Figure 8 sensors-22-09342-f008:**
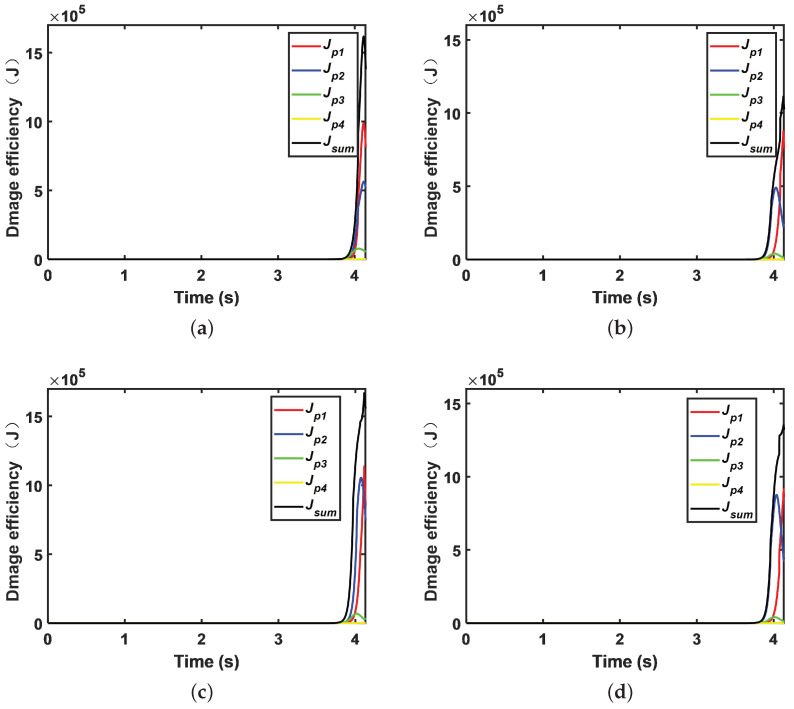
Unit fragment damage efficiency: (**a**) proposed strategy; (**b**) strategy from the literature [13]; (**c**) strategy of Equation (Equation 24) with high precision; and (**d**) strategy of Equation (Equation 24) with low precision.

**Figure 9 sensors-22-09342-f009:**
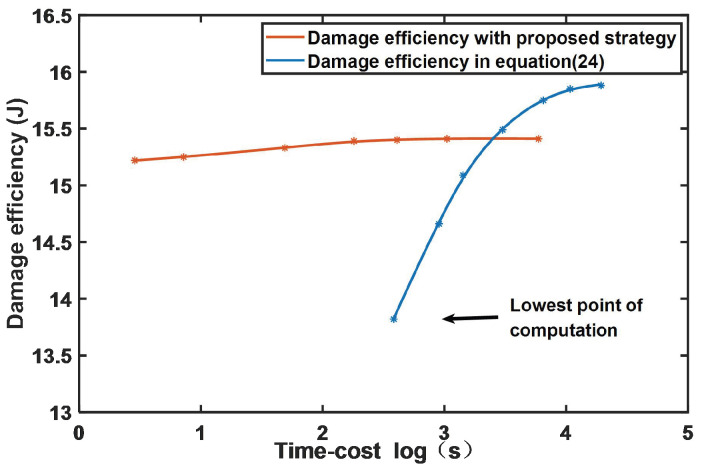
Damage efficiency under different time-costs.

**Figure 10 sensors-22-09342-f010:**
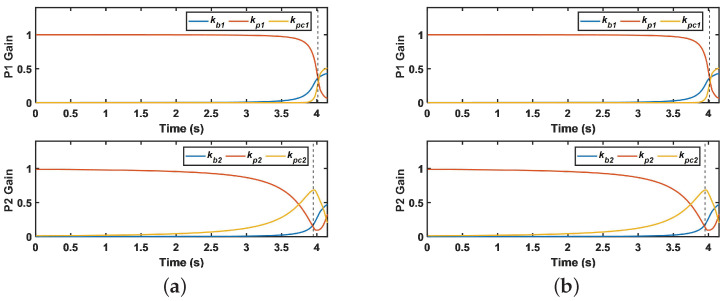
Gain of pursuer: (**a**) gain of $p_{1}$ and $p_{2}$ and (**b**) gain of $p_{3}$ and $p_{4}$.

**Figure 11 sensors-22-09342-f011:**
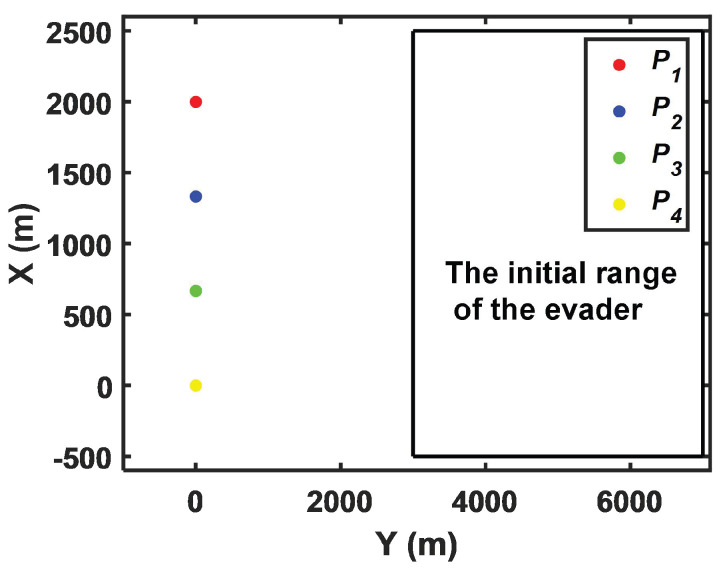
The initial range of the evader.

**Figure 12 sensors-22-09342-f012:**
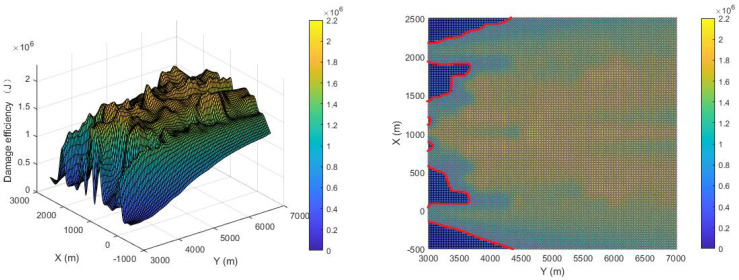
Damage efficiency of proposed strategy.

**Figure 13 sensors-22-09342-f013:**
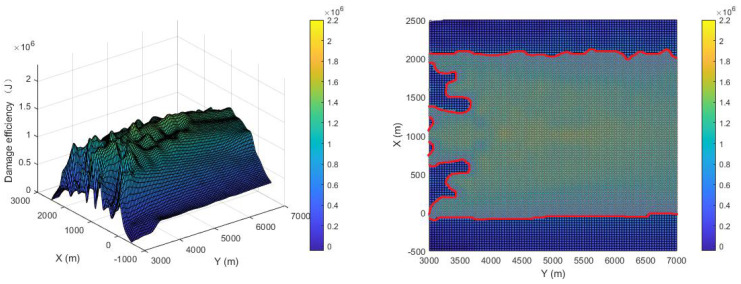
Damage efficiency of [13].

**Figure 14 sensors-22-09342-f014:**
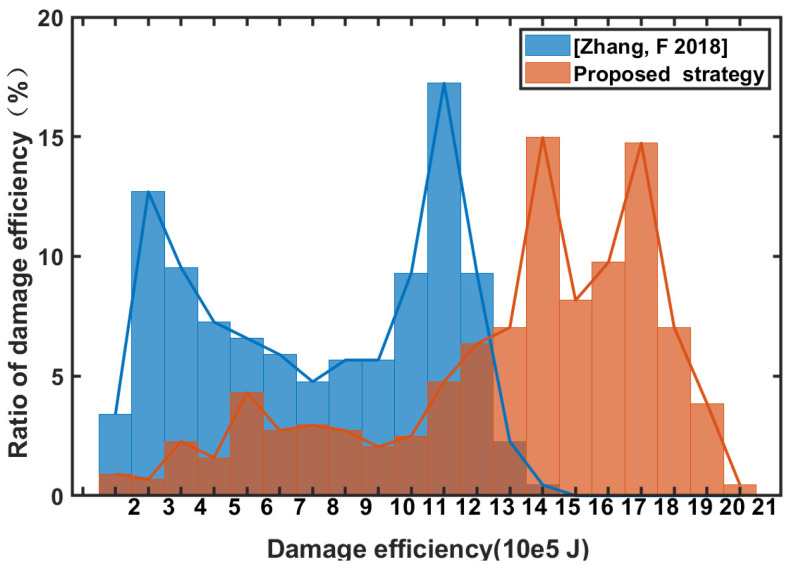
Histogram of the proportional distribution of damage efficiency [13].

**Figure 15 sensors-22-09342-f015:**
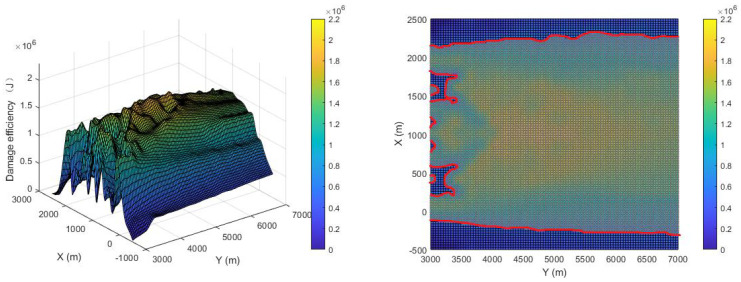
Damage efficiency in [13].

**Figure 16 sensors-22-09342-f016:**
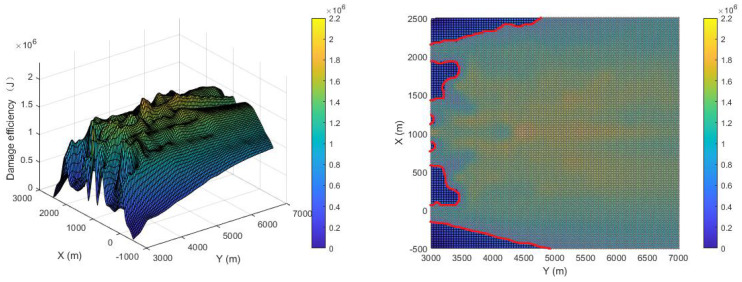
Damage efficiency in [13].

**Figure 17 sensors-22-09342-f017:**
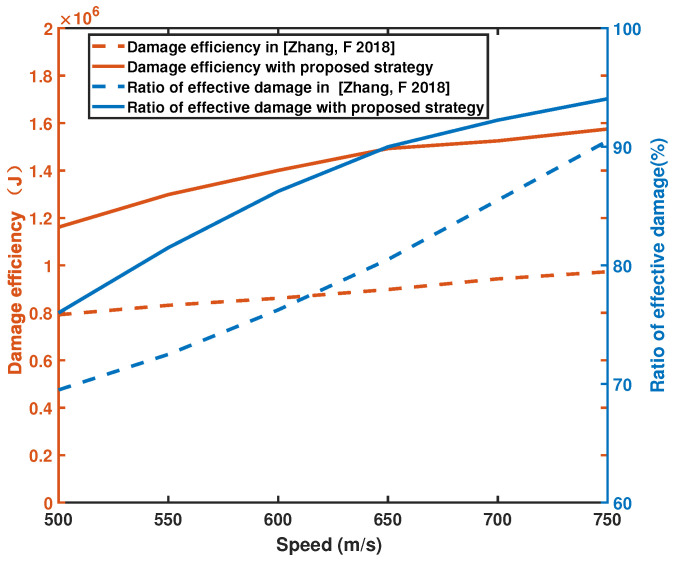
Monte Carlo simulation of different pursuer speeds [13].

**Table 1 sensors-22-09342-t001:** Participants’ initial parameter settings.

Participant	*X*/m	*Y*/m	Initial Angle/°	Speed/m·s${}^{-1}$
$p_{1}$	0	2000	0	600
$p_{2}$	0	1333	0	600
$p_{3}$	0	667	0	600
$p_{4}$	0	0	0	600
*e*	60,000	1200	90	800

**Table 2 sensors-22-09342-t002:** Damage efficiency model.

Parameter	Value	Parameter	Value
Gurney coefficient	2370	Correction coefficient	1.1
Detonation velocity of explosive	6930	Explosive loading factor	1
Damage range angle	[−15, 15]	Velocity attenuation coefficient	100

## Data Availability

Not applicable.

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
