# Peer review of "Optimal-Damage-Effectiveness Cooperative-Control Strategy for the Pursuit–Evasion Problem with Multiple Guided Missiles"

_sensors, 2022, doi:10.3390/s22239342_

Round 1

Reviewer 1 Report

This paper investigates the damage optimal cooperative control strategy based on a damage efficiency model and virtual force method and virtual force method. This paper is overally well- organized and some of the results seem interesting. The reviewer has the following concerns:

(1) Why do you utilize (1) as the dynamic mode? It seems to the author that it is not the fully form of an aircraft or a mobile robot since only kinematic is considered. Can be velocity be changing or actively controlled?
(2) What are the differences between the assumptions in Section 2.1 (3) and conventional guidance systems? Can the advantages of the system proposed in this paper be specified?

(3) Why is the range of theta_e distinguished in Formula (27)? What are the practical implications of this?

(4) The real time performance of Algorithm 1 needs to be discussed and simulated.

(5) Any qua quantitative between the current submission and the existing results?

(6) The authors are recommended to proofread this submission carefully in the revised version.

Reviewer 2 Report

Aiming at the pursuit-avoidance problem of multiple missiles, a damage-optimized-efficiency cooperative control strategy based on damage efficiency model and virtual force method is proposed. This is an interesting paper, very good. However, it needs some improvement.

(1) What are the incomparable advantages of the "virtual force" method in terms of computational complexity? I recommend that the authors explain clearly why they prefer to use the "virtual force" approach.

(2) Please add a new formulation of "virtual force" in the introduction.

(3) Please provide context for the findings presented in the framework of the introduction so that scientists working outside the topic of the paper can understand the introduction.

(4) Is the proposed approach applicable to other recovery evasion issues?

(5) The authors need more discussion on the limitations of the proposed model.

(6) Future works can be postponed. 

Reviewer 3 Report

In this paper, a damage optimal effectiveness cooperative control strategy based on the damage efficiency model and virtual force method is proposed to solve the pursuit-evasion problem of multiple guided missiles. Here are some concerns which should be well addressed when preparing a revised version.

(i) The paper has some typos which need to be checked. For example, there should be a space after the ‘pursuer;’ in lines 77 and 81 ; Principle should be changed to principle in line165 ; Figure3 in line 204 should be changed to Figure 3. In line 261, Figure ?? is a misquotation. Authors should check the full text for similar errors and make corrections.

(ii) The format of the references cited in the article is not uniform. For example, q, y, g should be changed Q, Y, G in reference 4; The title of the paper only needs to capitalize the first letter of the first word; The name of the journal should be IEEE Transactions on Automatic Control in reference 6.

(iii)Authors should add a detailed remark to compare the results in this paper with those of recent studies.
